# LEAGUE: LEADERBOARD GENERATION ON DEMAND

## ABSTRACT

Leaderboard gathers experimental results from various sources into unified rankings, giving researchers clear standards for measuring progress while facilitating fair comparisons. However, with thousands of academic papers updated daily, manually tracking each paper's methods, results, and experimental settings has become impractical, creating an urgent need for automated leaderboard generation. Although large language models offer promise in automating this process, challenges such as multi-document summarization, fair result extraction, and consistent experimental comparison remain underexplored. To address these challenges, we introduce Leaderboard Auto Generation (League), a novel and well-organized framework for automatic generation of leaderboards on a given research topic in rapidly evolving fields like Artificial Intelligence. League employs a systematic pipeline encompassing paper collection, result extraction and integration, leaderboard generation, and quality evaluation. Through extensive experiments across multiple research domains, we demonstrate that League produces leaderboards comparable to manual curation while significantly reducing human effort. [1]

## 1 INTRODUCTION

The explosive growth of scientific publications has created both unprecedented opportunities and significant challenges for researchers seeking to stay abreast of state-of-the-art methods (Bornmann et al., 2020; Wang et al., 2024; Şahinuç et al., 2024). Leaderboard platforms, such as NLP-progress[2] and Papers-With-Code[3] have become invaluable by offering comprehensive overviews of recent research developments, highlighting ongoing trends, and identifying future directions. However, the overwhelming growth of daily papers makes it increasingly difficult to build and update leaderboards automatically and promptly. Figure 1 illustrates two pressing issues: 1) a widening gap: LLM-related paper submissions to arXiv have surged year-over-year (exceeding 55,000 in 2025), yet the growth trend of leaderboard submissions on Paper with Code (5670 in 2025) has not advanced accordingly. 2) Even as new methods continuously emerge, leaderboards, such as the one for Multi-hop Question Answering on the HotpotQA(Yang et al., 2018) dataset, remain stagnant, with the latest method dating back to 2023. These observations highlight a serious challenge: the rapid accumulation of daily scientific publications often outpaces the capability of researchers to keep up with cutting-edge research and state-of-the-art methods, emphasizing the growing need for more efficient methods to generate the latest and useful leaderboards.

Earlier studies have made initial attempts to tackle this challenge. Research on scientific information extraction (Hou et al., 2019; Kardas et al., 2020) has focused on identifying entities such as models, datasets, metrics, and results from individual NLP papers, which enables the creation of paper-specific leaderboards. However, such approaches are inherently difficult to maintain and update over time. More recently, Li et al. (2023), proposed Scientific NER to extract entities from both text and tables, while Şahinuç et al. (2024) introduced SCILEAD, a hand-curated dataset containing 27 leaderboards from 43 NLP papers for evaluating LLMs on entity extraction. Despite these advances, prior methods remain restricted to entity extraction, offering only preliminary building blocks for leaderboard creation and yielding static snapshots from a narrow set of papers. In contrast, our work shifts the focus to **dynamic leaderboard construction**, where the goal is not only to extract results but also to continuously track, align, and compare model performance on specific datasets or tasks

---

[1]Code will be available upon publication.
[2]https://nlpprogress.com/
[3]https://paperswithcode.com/

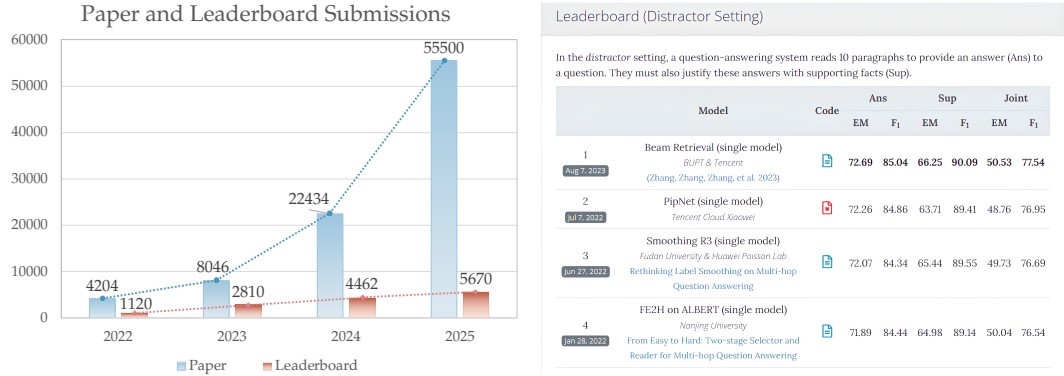

Figure 1: Left: Growth trend of paper and leaderboard submissions on LLMs from 2022 to 2025-09. The leaderboard statistics are collected from Paper with Code. Right: Leaderboard of Multi-hop QA, the latest method is still stuck in 2023.

under standardized evaluation settings. This enables researchers to observe evolving performance trends and conduct fair comparisons, thereby providing a more comprehensive and up-to-date view of progress in a research area.

Directly applying LLMs such as GPT-4 (Achiam et al., 2023), Qwen (Yang et al., 2024), and O1-preview that have demonstrated exceptional performance in various NLP tasks, especially in the long-context scenario (Chen et al., 2023a;b; Wang et al., 2023b) to this task faces several key challenges. First, **Limited Paper Coverage**: It is challenging for humans to search for all papers on a certain scientific topic, due to the overwhelming number of constantly emerging publications. Second, **Unfair Comparison**: Current studies do not consider fair experiment settings when making comparisons. For example, in NLP research, key experimental components, model size, train dataset size, and hyperparameter selection vary significantly across publications, highlighting the need for automatic alignment. Finally, **Low Timeliness**: A leaderboard, which lacks regular updates and continuous maintenance, cannot provide researchers with sufficient useful information.

To address these issues, we introduce League, a novel framework for dynamically and automatically leaderboard generation. Figure 2 illustrates the framework of our method, which is organized into four stages: (1) **Paper Collection and split**: Initially, League automatically downloads all relevant LaTex and PDF files based on the given research topic from arXiv and top-tier conferences, including ACL, EMNLP, NeurIPS, ICML, and ICLR, and filters out papers published before a certain date and those unrelated to the topic, ensuring proper paper coverage and timeliness. (2) **Table Extraction and Classification**: We use LLMs to extract and classify experiment tables based on accompanying table descriptions. (3) **Table Unpacking and Integration**: League extracts the datasets, metrics, experiment settings, and experiment results from the tables in the form of a quintuple, including paper titles. Experiment setting extraction is crucial for enabling fair comparisons across different baselines (including model size, data size, etc). (4) **Leaderboard Generation and Evaluation**: The extracted quintuples are recombined, refined, and re-ranked to form candidate leaderboards.

We assess performance along two dimensions: (1) Topic-related Quality: whether each quintuple in League-generated leaderboards relates to the given topic; (2) Content Quality: evaluated via LLM-as-Judge on four aspects including Coverage, Structure, Latest, and Multiaspect. Human experts also evaluate the leaderboards, with Pearson Correlation computed between human and LLM scores. Experiments across different leaderboard lengths (5, 10, 15, and 20 items) show that League consistently achieves high Topic-related and Content Quality scores, approaching human performance but about 5-10 times the speed of human annotation on 20-item leaderboard construction. With 20 items, a League-generated Leaderboard represents 20 baselines for researchers, achieving 67.58% recall and 70.33% precision scores in Topic-related Quality. In Content Quality with 20 items, League achieves 4.12 coverage, 3.96 latest, 4.16 structure, and 4.08 multiaspect scores, approaching human performance (4.72 coverage, 4.68 latest, 4.34 structure, and 4.58 multiaspect scores). Although the manually created leaderboard achieves higher Content Quality, it is much more time-consuming than League, highlighting League's superior efficiency. With fewer items, League gets even higher performance, slightly lower than human performance. These results highlight the effectiveness of League, providing a reliable proxy for human judgment across varying leaderboard

Table 1: Comparison between related work and our approach. **Data Source**: Source of leaderboards. NProg.: *NLP-progress*, PwC: *paperswithcode*. **Experiment Settings**: whether the experiment settings are extracted as part of leaderboards. **Multi Document**: whether the leaderboards are constructed from multiple papers. **Dynamic**: whether the generated leaderboards can be updated dynamically.

| Related Work | Data Source | Exp. Settings | Multi Doc. | Dynamic | Timeliness |
|---|---|---|---|---|---|
| TDMS(Hou et al., 2019) | N Prog. | ✗ | ✗ | ✗ | ✗ |
| Axcell (Kardas et al., 2020) | PwC | ✗ | ✗ | ✗ | ✗ |
| TELIN (Yang et al., 2022) | PwC | ✗ | ✗ | ✗ | ✗ |
| ORKG(KABENAMUALU et al., 2023) | PwC | ✗ | ✗ | ✗ | ✗ |
| LEGO (Singh et al., 2024) | PwC | ✗ | ✗ | ✗ | ✗ |
| SciLead (Şahinuç et al., 2024) | NLP papers | ✗ | ✔ | ✗ | ✗ |
| League (Ours) | arXiv | ✔ | ✔ | ✔ | ✔ |

items. Furthermore, the Pearson Correlation Coefficient values indicate a moderate positive correlation between the human-assigned and LLM-assigned scores. To the best of our knowledge, we are the first to explore the potential of LLMs for automatic leaderboard generation, proposing evaluation criteria that align with human preferences and offering valuable reference for future related research.

## 2 RELATED WORK

**LLM for Scientific Research.** Several studies have explored using LLMs to improve work efficiency in scientific research (Xie et al., 2025). Baek et al. (2024) and Yang et al. (2023) propose a multi-agent-based scientific idea generation method to boost AI-related research. To evaluate the quality of LLM-generated ideas, Si et al. (2024) introduces a comprehensive human evaluation metric. Wang et al. (2023a) proposes SciMON, a method that uses LLMs for retrieving the scientific literature. Wang et al. (2024) proposes an AutoSurvey to automatically generate scientific surveys based on the given research topic. The AI Scientist (Lu et al., 2024) introduces a fully automated and prompt-driven research pipeline. To make LLM-generated ideas more diverse and practical, Weng et al. (2024) proposes an iterative self-rewarding framework that allows the LLM to continuously refine its ideas, improving both diversity and practicality in research proposal generation. However, no previous research focused on leaderboard generation for researchers to search, organize, and compare the state-of-the-art methods rapidly and fairly based on a certain research topic.

**Scientific Information Extraction.** Table 1 illustrates the differences between the previous experiment results extraction work and League. Earlier works on scientific information extraction mainly focused on extracting entities such as Task, Dataset, and Model (TDM triples) from sources like NLP-progress or Papers-with-Code (Hou et al., 2019; Kardas et al., 2020; Singh et al., 2024). Others (Yang et al., 2022; KABENAMUALU et al., 2023) extended this approach by leveraging predefined TDM taxonomies. However, these methods face three key limitations: (1) they rely on incomplete or inconsistently curated sources, leading to gaps in coverage; (2) they depend on fixed taxonomies, making them inflexible to new benchmarks or emerging research directions; and (3) they only extract entities without capturing experimental settings, preventing fair and reproducible comparisons. As a result, prior efforts can at best provide static snapshots of research progress, but fail to maintain up-to-date, comparable leaderboards.

In contrast, our League framework goes beyond mere extraction: it automatically crawls arXiv and top conference proceedings, identifies and unpacks main experimental tables, extracts both results and experiment settings, and then integrates these quintuples to produce complete, up-to-date leaderboards. By combining dynamic paper collection, table classification, settings-aware result integration, and LLM-based refinement and evaluation, League generates leaderboards end-to-end rather than only extracting isolated result entities. This shift enables timely, reproducible, and fair comparisons across methods and makes League suitable for rapidly evolving research areas where static extraction methods fall short.

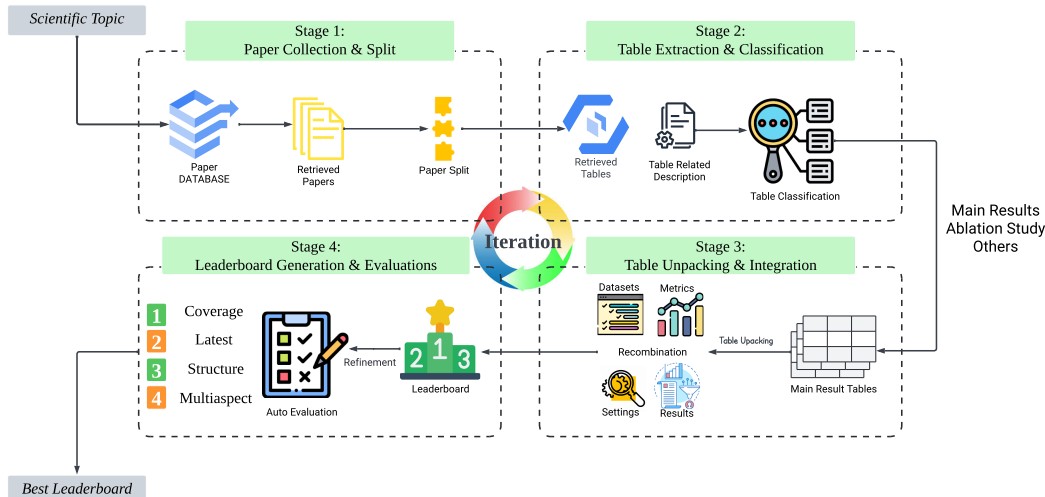

Figure 2: The League framework for leaderboard automatic generation. In Stage 1, we automatically crawl scientific papers from arXiv. In Stage 2, we retrieve, extract, and classify tables from the LaTeX code. In Stage 3, we select the main results tables and extract datasets, metrics, results, and experiment settings from the main results table. In Stage 4, we generate Leaderboards from the selected results and evaluate the quality.

# 3 METHOD

Figure 2 depicts the proposed League, which consists of four stages: (1) Paper Collection and Split, (2) Table Extraction and Classification, (3) Table Unpacking and Integration, and (4) Leaderboard Generation and Evaluation. Each stage is meticulously designed to address specific challenges associated with leaderboard generation, thereby enhancing the efficiency and quality of the resulting leaderboards. The whole process is iterated several times (e.g., five times) to generate a high-quality leaderboard.

## 3.1 STAGE I PAPER COLLECTION AND SPLIT

Utilizing the off-the-shelf tools[4], League first searches and retrieves a set of papers $P_{\text{init}} = \{P_1, P_2, ..., P_N\}$ from arXiv and downloads LaTeX code files related to a specific research topic $T$. Then, we specify a certain date and filter out all papers published before the date. The filtering stage is important for ensuring that the generated leaderboards are grounded in the most relevant and recent research. Moreover, since the search tool just identifies only the keywords in the paper title and abstract, which can lead to a significant amount of noisy data, we also introduce a retrieval model to filter out papers that are irrelevant to the given topic and retrieve topic-related papers. The set of filtered papers $P_{\text{filtered}} = \{\text{Retrieval}\{P_1, P_2, ..., P_U\}\}$ is used to generate the leaderboards, ensuring comprehensive coverage of the topic and logical structure. Due to the extensive number of relevant papers retrieved and filtered during this stage, the total input length of $P_{\text{filtered}}$ often exceeds the maximum input length of LLMs. Since most of the LaTeX content is unproductive for generating leaderboards, we split the LaTeX code into several sections based on the structure of each paper. Most tables, table-related descriptions, experiment results and experiment settings are located in the "Experiment" section, which contains the key information to generate leaderboards. Thus, we select all "Experiment" sections as well as all tables $\{\text{Table}_1, \text{Table}_2, ..., \text{Table}_U\}$ and all table-related descriptions $\{D_1, D_2, ...D_U\}$, extracted from all papers, as input for the next stage.

## 3.2 STAGE II TABLE EXTRACTION AND CLASSIFICATION

Typically, a scientific paper, such as those in the natural language processing domain, contains several types of tables, including "Main Results", "Ablation Study", and "Others". The "Main Results"

---

[4]https://github.com/lukasschwab/arxiv.py

tables are the most important tables in the paper, which illustrate the novelty, contributions, and effectiveness of the proposed methods or models by comparing the experiment results of the proposed method with other baselines. We utilize these tables for leaderboard generation. The "Ablation Study" tables examine the effect of damaging or removing certain components in a controlled setting to investigate all possible outcomes of system failure. The "Others" tables are the tables that illustrate the supplementary information of the experiments. For example, some tables illustrate the dataset statistics of the benchmark used in the experiments, while other tables list the results of the "Case Study" and "Error Analysis". To address this, we propose a framework that uses the In-Context Learning method (Dong et al., 2022) to manually select one table from each of the three different types. The framework then prompts LLMs to classify the table types and only keeps the "Main Experiments" tables and their descriptions as the final input. The $i_{th}$ table types can be described as: $\text{LLM}(\text{Table}_i, D_i; \text{Prompt}) \to \texttt{Table type}$.

In practice, the most intrinsic approach is to divide Stage 2 into the following sequential steps: (1) Extract all tables and their associated captions from the LaTeX code. (2) Classify the extracted tables according to predefined table types. (3) Extract metrics, performance values, and experimental settings related to the proposed model from tables categorized as "Main Results". However, each of these three steps necessitates the use of LLM APIs, and repeated reference to certain table contents further exacerbates the substantial waste of tokens. To address this issue, we follow the few-shot Chain of Thought (CoT) prompting process, enabling it to classify and extract information from identified "Main Results" tables in a single dialogue round. Specifically, in the requested JSON output, we additionally set the key points as follows: "number of tables (Int)", "classification of tables (Dict)", and "selected table's index (Int)".

## 3.3 STAGE III TABLE UNPACKING AND INTEGRATION

Following the table extraction and classification phase, each table Table$_i$ is sent into the LLM to extract the core information. To build a useful and high-quality leaderboard, we define four types of scientific terms: Datasets, Metrics, Experiment Results, and Experiment Settings. We follow the definition of scientific entities proposed by SciIE (Li et al., 2023). For datasets, we use LLMs to count the frequency in all filtered papers $P_{\text{filtered}}$ of each dataset under a certain research topic and retain the top-K (K=5) datasets with the highest frequency of occurrence in scientific papers, ensuring the generated leaderboard contains enough methods. For the rest of the three scientific terms, we utilize LLMs to extract from given Table$_i$ with a related table description $D_i$. After scientific term extraction, we recombine them into a quintuple, including the paper title as the unique identification ID. Each paper can produce one quintuple, and finally, we get a raw leaderboard with $K * M$ quintuples from $M$ filtered papers and $K$ datasets. The raw leaderboard is then reranked on the basis of the experiment results.

## 3.4 STAGE IV LEADERBOARD GENERATION AND EVALUATION

Following the table unpacking and integration phase, we get $K * M$ quintuples, and all quintuples are formatted into $K$ leaderboards. Each leaderboard is individually refined by a third-party LLM such as GPT-4 to enhance readability, eliminate redundancies, and ensure completeness. After we obtain $K$ leaderboards, the final stage involves a quality evaluation based on our pre-defined four criteria, which is shown in Appendix Table 9. Each leaderboard is assigned three scores based on "Coverage", "Latest" and "Structure". Since a research topic may contain several datasets, the "Multi-Aspect" is the average quality score used to evaluate the LLM-generated leaderboards for each dataset. The best leaderboard is chosen from $N$ candidates. LLMs critically examine the leaderboards in several aspects. The final output of League is $L_{\text{best}} = \text{Evaluate}(L_{\text{ca1}}, L_{\text{ca2}}, ..., L_{\text{caN}})$. The methodology outlined here, from paper collection to leaderboard evaluation, ensures that League effectively addresses the complexities of leaderboard generation in the AI domain using advanced LLMs. We provide Pseudo-code for easily understanding, which is shown in Appendix Algorithm 1.

Table 2: Results of leaderboard quality generated by League in the first iteration (The results of the first iteration could help reflect the effectiveness and efficiency of our framework). **Items**: The number of items in the leaderboard. For example, a 5-item leaderboard contains 5 baselines. **Topic-related Quality**: The precision and recall of each paper in relation to its relevance to the topic. **Speed**: The average time required to generate a single leaderboard. **Content Quality**: The evaluation results of the leaderboard content. The best results of Leagure are highlighted in **bold**, the fastest results are underlined.

| Items | Topic-related Quality | | Model | Speed$_{/s}$ | Content Quality | | | |
| | Recall↑ | Precision↑ | | | Coverage↑ | Latest↑ | Structure↑ | Multiaspect↑ |
|---|---|---|---|---|---|---|---|---|
| 5 | $76.57_{\pm 11.65}$ | $79.43_{\pm 8.86}$ | Qwen2.5-7B | 131.43 | $3.60_{\pm 0.48}$ | $3.46_{\pm 0.49}$ | $3.18_{\pm 0.32}$ | 3.41 |
| | | | Qwen2.5-14B | 129.51 | $4.23_{\pm 0.38}$ | $4.14_{\pm 0.31}$ | $3.68_{\pm 0.29}$ | 4.02 |
| | | | GPT-4o | 49.64 | $4.52_{\pm 0.42}$ | $4.70_{\pm 0.32}$ | $4.32_{\pm 0.38}$ | 4.41 |
| | | | O1-preview | 79.67 | $\mathbf{4.63_{\pm 0.48}}$ | $\mathbf{4.71_{\pm 0.71}}$ | $\mathbf{4.40_{\pm 0.33}}$ | **4.58** |
| | | | Human Writing | 355 | 4.89 | 4.83 | 4.91 | 4.88 |
| 10 | $75.19_{\pm 9.81}$ | $80.05_{\pm 6.76}$ | Qwen2.5-7B | 156.41 | $3.22_{\pm 0.48}$ | $3.41_{\pm 0.49}$ | $4.11_{\pm 0.39}$ | 3.57 |
| | | | Qwen2.5-14B | 163.54 | $3.91_{\pm 0.48}$ | $3.55_{\pm 0.49}$ | $3.41_{\pm 0.39}$ | 4.61 |
| | | | GPT-4o | 88.96 | $\mathbf{4.68_{\pm 0.39}}$ | $\mathbf{4.59_{\pm 0.33}}$ | $\mathbf{4.45_{\pm 0.41}}$ | **4.56** |
| | | | O1-preview | 98.44 | $4.40_{\pm 0.48}$ | $4.46_{\pm 0.71}$ | $4.31_{\pm 0.33}$ | 4.39 |
| | | | Human Writing | 612 | 4.81 | 4.72 | 4.65 | 4.72 |
| 15 | $71.34_{\pm 8.39}$ | $74.58_{\pm 7.35}$ | Qwen2.5-7B | 183.45 | $3.11_{\pm 0.28}$ | $3.23_{\pm 0.26}$ | $3.15_{\pm 0.27}$ | 3.16 |
| | | | Qwen2.5-14B | 195.63 | $3.68_{\pm 0.28}$ | $3.32_{\pm 0.19}$ | $3.18_{\pm 0.24}$ | 3.39 |
| | | | GPT-4o | 105.61 | $\mathbf{4.47_{\pm 0.22}}$ | $\mathbf{4.16_{\pm 0.27}}$ | $\mathbf{4.32_{\pm 0.24}}$ | **4.28** |
| | | | O1-preview | 109.33 | $4.21_{\pm 0.48}$ | $4.06_{\pm 0.21}$ | $4.28_{\pm 0.31}$ | 4.18 |
| | | | Human Writing | 839 | 4.71 | 4.65 | 4.44 | 4.60 |
| 20 | $67.58_{\pm 9.12}$ | $70.33_{\pm 6.89}$ | Qwen2.5-7B | 196.33 | $3.03_{\pm 0.25}$ | $3.11_{\pm 0.31}$ | $2.98_{\pm 0.25}$ | 3.16 |
| | | | Qwen2.5-14B | 208.64 | $3.49_{\pm 0.34}$ | $3.17_{\pm 0.26}$ | $3.03_{\pm 0.28}$ | 3.39 |
| | | | GPT-4o | 120.52 | $\mathbf{4.28_{\pm 0.28}}$ | $3.92_{\pm 0.22}$ | $\mathbf{4.21_{\pm 0.25}}$ | **4.13** |
| | | | O1-preview | 117.45 | $4.12_{\pm 0.38}$ | $\mathbf{3.96_{\pm 0.25}}$ | $4.16_{\pm 0.29}$ | 4.08 |
| | | | Human Writing | 1128 | 4.72 | 4.68 | 4.34 | 4.58 |

## 4 EXPERIMENTS

We designed experiments for League, with the aim of answering four questions: RQ-1: Can League address the paper coverage issue and generate fair leaderboards by incorporating the latest baselines? RQ-2: Can League reduce time consumption compared to human? RQ-3: Is the evaluation consistent between League and human experts? RQ-4: Is each proposed component of League useful?

### 4.1 EXPERIMENTAL SETUP

#### 4.1.1 EVALUATION METRICS

We use two metrics to evaluate the quality (topic-related and leaderboard content) and the speed of leaderboard generation, respectively.

Table 3: Left: Table Classification result on the extracted tables using different LLMs. Right: Table NER result on the manually annotated table entities.

| Methods | Prompt | P (%) | R (%) | F1 (%) |
|---|---|---|---|---|
| GPT-4o | 0-shot | 80.96 | 82.49 | 80.03 |
| | 1-shot | 87.45 (+6.49) | 86.44 (+3.95) | 85.10 (+5.07) |
| O1-preview | 0-shot | 81.16 | 80.23 | 78.55 |
| | 1-shot | 85.19 (+4.03) | 83.62 (+3.39) | 82.78 (+4.23) |

| Methods | Prompt | P (%) | R (%) | F1 (%) |
|---|---|---|---|---|
| GPT-4o | 0-shot | 86.51 | 87.95 | 85.30 |
| | 1-shot | 89.17 (+2.66) | 90.26 (+2.31) | 88.76 (+3.46) |
| O1-preview | 0-shot | 84.28 | 85.44 | 83.76 |
| | 1-shot | 88.55 (+4.27) | 90.02 (+4.58) | 89.82 (+6.06) |

**(1) Topic-related Quality:** The aforementioned arXiv crawler employs regular expression matching in the abstract section to identify papers related to specific topics. While this method is efficient, it is relatively rudimentary and cannot guarantee that all retrieved papers meet our requirements. The quality of these papers not only directly affects the final leaderboard, but low-quality candidate papers can also significantly prolong the time required for construction. Therefore, it is essential to evaluate the quality of the retrieved articles. We evaluate the quality of content from the following two aspects. **(i) Recall:** It measures whether all items in the generated leaderboard are related to the given research topic. **(ii) Precision:** It identifies irrelevant items, ensuring that the items in the generated leaderboards are pertinent and directly support the given research topic.

**(2) Leaderboard Content Quality:** The evaluation metric of leaderboard Content Quality includes four aspects. Each aspect is judged by LLMs according to a 5-point, calibrated by human experts. The evaluation criteria are listed in Appendix Table 9. **(i) Coverage**: Each paper represented on the League-generated leaderboards encapsulates all aspects of the topic. **(ii) Latest**: Test whether all papers represented on the League-generated leaderboards are the latest. **(iii) Structure**: Evaluate the logical organization and determine whether the League leaderboards have missing items. **(iv) Multiaspect**: Average score of the previous three criteria for League-generated leaderboards.

**(3) Leaderboard Construction Speed:** Manually building a leaderboard is a time-consuming and laborious task. This process can be divided into the following main components: $T_r$ (search for papers on a specific topic), $T_b$ (browse all retrieved articles and develop several highly frequent data), $T_f$ (filter candidate papers based on the selected data), $T_e$ (read and extract information), and $T_c$ (the integration and construction time). The total time consumption is calculated as:

Given $L$ denotes the length of the leaderboard, $N_{\text{retrieved}}$ is the number of retrieved articles, $N_{\text{filtered}}$ is the number of articles retained, and $P$ is the proportion of valid articles with $P = \frac{N_{\text{filtered}}}{N_{\text{retrieved}}}$. We find that $T_b$ and $T_f$ are strongly correlated with leaderboard length $L$ and the Topic-related Quality:

$$\{T_b, T_f\} \propto \frac{L}{P} = \frac{L \cdot N_{\text{retrieved}}}{N_{\text{filtered}}}. \tag{1}$$

Although $T_r$ is relatively fixed, $T_e$ and $T_c$ usually only have a positive correlation with $L$.

For League, we directly account for all the invocation time of the LLMs' API calls. Compared to manual work, which often takes several days, League reduces the total time cost at the minute level. This is largely attributed to the task decomposition conducted in this paper, the division of labor and scheduling within the framework, and the superior performance of the LLMs.

### 4.1.2 BASELINES

We employ proprietary and open-source LLMs in our experiments and set the sampling temperature to 0.7 for proprietary models. For proprietary models, we adopt GPT-4o (Achiam et al., 2023), and the O1-preview. For open-source LLMs, we adopt Qwen2.5-7B and Qwen2.5-14B (Yang et al., 2024). We provide a detailed illustration of our prompts for different stages in Appendix C.

### 4.2 EXPERIMENT RESULTS

#### 4.2.1 PERFORMANCE COMPARISON (RQ-1)

**Topic-related Quality Evaluation**: Table 2 illustrates the Topic-related Quality League achieves a recall of 67.58% and a precision of 70.33% with 20 items, indicating that it successfully retrieves a large proportion of relevant papers while maintaining a low rate of irrelevant ones. The high precision and recall scores show that League can help solve the paper coverage problem. This performance is crucial to ensure that the generated leaderboards are both comprehensive and accurate.

**Fair Comparison**: To ensure a fair comparison, League extracts all experiment settings from crawled papers as part of the League-generated leaderboards. We provide a detailed case study of League-generated leaderboards with experiment settings in Appendix E.

**Content Quality Evaluation**: Table 2 and Table 5 in appendix present the results of the quality of the leaderboard generated by League. Items in Table 2 are crawled from arXiv while items in Table 5 are from the top-tier conferences (including ACL, EMNLP, NeurIPS, ICML, ICLR, etc.) respectively, which shows that League could guarantee the timelines and quality of the generated leaderboards. The papers crawled from arXiv could provide the latest research trend on the given topic, while papers crawled from top conferences could help researchers build leaderboards with higher quality, compared with the Topic-related Quality and Content Quality in Table 2 and Appendix Table 5. League consistently achieves high scores across all evaluation metrics, particularly in terms of Coverage and Latest, indicating its ability to include a wide range of relevant and recent papers. For example, at a leaderboard length of 20 items, League achieves a Coverage score of 4.12 and a Latest score of 3.96, approaching human performance (4.72 and 4.68, respectively). While manual leaderboards score slightly higher in Content Quality, League significantly reduces the time required for leaderboard generation from 1128s to 117.45s, demonstrating its efficiency. We also list

a 5 item leaderboard generated by League, shown in Figure 3, which offers significant advantages over the outdated official HotpotQA leaderboard (Figure 1). Our method shows three advantages: 1) **Latest**, all papers are published after 2024; 2) **High Quality**, all papers are collected from top-tier conferences, including ACL, EMNLP, ICLR, and ICML; 3) **More Information**, all items on the leader board contain experiment details, including models and training strategies.

Table 4: Ablation study for League with components removed. We use the GPT-4o as the backbone.

| Methods | Items | Speed$_{/s}$ | Coverage↑ | Latest↑ | Structure↑ | Multiaspect↑ |
|---|---|---|---|---|---|---|
| | | | | | Content Quality | |
| League *w/o* Table Classification | 5 | 42.35 | 4.43 | 4.52 | 3.95 | 4.30 |
| League *w/o* Refinement | | 43.58 | 4.41 | 4.46 | 4.05 | 4.31 |
| League | | 49.64 | **4.52** | **4.70** | **4.32** | **4.51** |
| League *w/o* Table Classification | 10 | 81.37 | 4.31 | 4.36 | 4.01 | 4.33 |
| League *w/o* Refinement | | 80.52 | 4.24 | 4.17 | 3.88 | 4.10 |
| League | | 88.96 | **4.68** | **4.59** | **4.45** | **4.56** |
| League *w/o* Table Classification | 15 | 93.28 | 4.13 | 4.08 | 3.61 | 3.94 |
| League *w/o* Refinement | | 91.32 | 4.19 | 4.13 | 3.72 | 4.01 |
| League | | 105.61 | **4.47** | **4.16** | **4.32** | **4.28** |
| League *w/o* Table Classification | 20 | 105.31 | 3.92 | 3.88 | 3.51 | 3.77 |
| League *w/o* Refinement | | 99.35 | 3.85 | 3.71 | 3.62 | 3.72 |
| League | | 120.52 | **4.28** | **3.92** | **4.21** | **4.13** |

**Iteration Evaluation**: To ensure the high-quality of League-generated leaderboards, we iterate the process to evaluate the performance change during the whole iteration. The left part of Appendix Figure 4 presents the effect of different iteration counts on the performance of League. The results show that increasing the number of iterations from 1 to 5 provides a significant improvement in Structure quality and Coverage quality scores. The Latest score remains at a relatively high level, which is because in stage 1 of the League, the old papers are filtered out. To sum up, our experiments demonstrate that League is highly effective in generating high-quality, up-to-date leaderboards across various research topics. The framework's ability to dynamically update leaderboards and extract detailed experiment settings ensures a fair comparison between state-of-the-art baselines. While League's Content Quality scores are slightly lower than those of manually created leaderboards, its efficiency and scalability make it a valuable tool for researchers in rapidly evolving fields like AI and NLP.

### 4.2.2 EFFICIENCY ANALYSIS (RQ-2)

**Construction Speed**: League dramatically reduces the time required to generate leaderboards compared to manual methods. For instance, generating a 20-item leaderboard with League takes approximately 2 minutes, while manual construction takes more than 18 minutes. This speed advantage makes League a practical tool for researchers who need up-to-date leaderboards in rapidly evolving fields. The high speed of League shows that it can generate high-quality leaderboards timely.

### 4.2.3 META EVALUATION (RQ-3)

To verify the consistency between our proposed LLM evaluation strategy and human evaluation, we conduct a correlation evaluation involving human experts and our automated evaluation method. Human experts judge pairs of generated leaderboards to determine which one is superior. We compare the judgments made by our method against those made by human experts. Specifically, we provide experts with the same scoring criteria as used in our evaluation as a reference. Experts rank the 20 League-generated leaderboards and compare these rankings with those generated by LLM using the Pearson Correlation Coefficient to measure the consistency between human and LLM evaluations. The results of this meta-evaluation are presented in the right part of Appendix Figure 4. The table shows the Pearson Correlation Coefficient values, indicating a strong positive correlation between the quality scores provided by LLM and those given by human experts, with the O1-preview achieving the highest correlation at **0.76**. These results suggest that our evaluation aligns well with human preferences, providing a reliable proxy for human evaluation.

**Question Answering in LLMs Leaderboard: HotpotQA dataset (ACL, EMNLP, ICLR, ICML)**

Papers due: 2025 September

Latest 5 papers

| No. | Model Name | Code | Training Strategy | EM | F1 | Source |
|---|---|---|---|---|---|---|
| 1 | Open-RAG: Enhanced Retrieval Augmented Reasoning with Open-Source Large Language Models
Llama2-13B | GitHub | Hybrid Approach for Adaptive Retrieval | 66.2 | 80.1 | EMNLP 2024 findings |
| 2 | HOLMES: Hyper-Relational Knowledge Graphs for Mulit-hop Question Answering using LLMs
GPT4-1106-preview | — | Distractor Setting, Auxiliary Graph Schema Creation | 66.0 | 78.0 | ACL 2024 |
| 3 | SiReRAG: Indexing Similar and Related Information for Multihop Reasoning
GPT-3.5-Turbo | GitHub | Indexing Both Similar and Related Information | 62.50 | 77.36 | ICLR 2025 |
| 4 | Mitigating Lost-in-Retrieval Problems in Retrieval Augmented Multi-Hop Question Answering
GPT-4o-mini | GitHub | Sub-question Decomposition with Sentence Graph | 50.0 | 64.22 | ACL 2025 |
| 5 | In-Context Sharpness as Alerts: An Inner Representation Perspective for Hallucination Mitigation
Llama2-70B-chat | GitHub | Propose an Entropy-Based Metric to Qualify In-Context Sharpness | 31.2 | 30.2 | ICML 2025 |

Figure 3: The example leaderboard generated by League. Comparing with the Leaderboard of Multi-hop QA method in the right part of Figure 1, our method could help summarize the experiment results from the top conference papers with experiment settings for fair comparison.

### 4.2.4 INTERMEDIATE EVALUATION

To validate the effectiveness of League's intermediate stages, we conducted experiments focusing on table classification and table-level entity recognition, as summarized in Table 3. **Table Classification.** We divided tables into three categories: (i) main results, (ii) ablation studies, and (iii) others (e.g., dataset statistics or case studies). The results demonstrate that League is highly effective at isolating the correct "main results" tables required for leaderboard construction. Both GPT-4o and O1-preview provide strong performance (over 80% F1 scores respectively with 1-shot prompt). This suggests that even minimal supervision helps the model better distinguish between relevant and auxiliary tables, thereby reducing noise in the subsequent leaderboard generation stage. **Table NER.** Following Li et al. (2023), we cast entity extraction as a classification problem, requiring the model to identify four key scientific entities: methods, datasets, experimental settings, and metrics. On a manually annotated validation set, League shows robust performance across all categories, with notable improvements when enhanced prompting strategies are applied. These results confirm that the system can reliably extract structured information from experimental tables, which is critical for generating comparable high-quality leaderboards. Details of these tasks are shown in Appendix D.

### 4.2.5 ABLATION STUDY (RQ-4)

To understand the contribution of each component in League, we conduct an ablation study by removing key components of League as follows: (1) League w/o Table Classification: We remove the table classification step, which leads to a slight decrease in Structure and Multiaspect scores, indicating that classifying tables is essential for maintaining a logical and well-organized leaderboard. (2) League w/o Refinement: We disable the Refinement step, which results in a minor drop in Coverage and Latest scores, suggesting that Refinement helps refine the leaderboard by ensuring that only the most relevant and recent papers are included. As shown in Table 4, the results confirm that each component plays a crucial role in achieving the generation of a high-quality leaderboard.

## 5 CONCLUSION

We introduced League, a novel framework leveraging LLMs to automatically generate the latest high-quality leaderboards based on given research topics. League addressed key challenges, including paper coverage, fair comparison, and timeliness, through a systematic approach that involves paper collection and splitting, table extraction and classification, table unpacking and integration, and leaderboard generation and evaluation. Experiments showed that League can automatically generate new high-quality leaderboards in a relatively short time and match human performance in terms of Topic-related Quality and Content Quality. This advancement offered a scalable and effective solution for synthesizing the latest leaderboards, providing a valuable tool for researchers in rapidly evolving fields such as artificial intelligence.

## ETHICS STATEMENT

All authors affirm their adherence to the ICLR Code of Ethics. We have carefully considered the ethical implications of our research, particularly concerning the safe and responsible deployment of Large Language Model (LLM)s. Our work directly addresses the critical need to avoid harm by mitigating risks such as dangerous diagnostic medical recommendations, financial losses, and privacy breaches, which can arise from the unconstrained operation of LLMs. We believe our work contributes positively to human well-being by enhancing the safety and trustworthiness of advanced AI systems.

## REPRODUCIBILITY STATEMENT

To ensure the reproducibility of our work, we have made significant efforts to document our methodology thoroughly. The full description and algorithm details of the League framework are described in Section 3 and algorithm 1. Our source code is provided in supplementary materials. We are committed to fostering open science and facilitating the replication of our results.

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

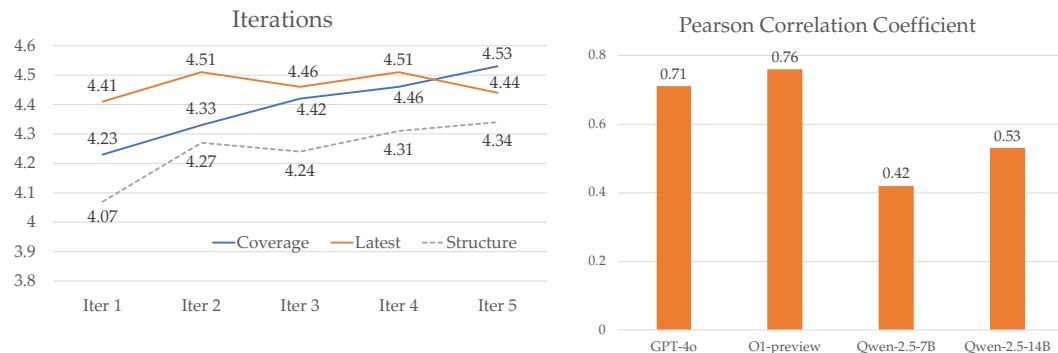

Figure 4: Left: Impact of Iteration on League Performance. Right: Pearson Correlation Coefficient values given by four LLMs and human experts. Note that the Pearson Correlation Coefficient is between -1 and 1, the larger value indicates more positive correlations.

# A    LIMITATIONS

One limitation of League is its reliance on the quality of the retrieved papers. While our Topic-related Quality metrics are strong, there is still room for improvement in ensuring that all relevant papers are included. Future work could explore more sophisticated retrieval models to further enhance the coverage of the generated leaderboards. Another limitation is that a specific dataset may contain several evaluation metrics, and different papers may use different metrics to evaluate proposed models' performance, bringing challenges for leaderboard generation and baseline comparison.

# B    THE USE OF LARGE LANGUAGE MODELS

We employed LLMs for grammar checking and polishing the English expression throughout this manuscript. It is important to note that while our research focuses on leveraging LLMs for automatic leaderboard generation, the LLMs studied in this work are the subject of our research rather than tools for research ideation or scientific writing. All experimental design, analysis, and scientific conclusions were developed independently by the authors.

---

**Algorithm 1** Leaderboard Automatic Generation.

---

1: **Input:** Scientific topic $T$, open-access platform arXiv $A$
2: **Output:** Final refined and evaluated leaderboard $L$
    *# Stage 1: Paper Collection and Document Split*
3: Crawl topic $T$ related $N$ publications $P_{\text{init}} = \{P_1, ...P_N\} \leftarrow \text{Retrieve}(T, A)$
4: Filter out topic-unrelated and old papers, $P_{\text{filtered}} = \{P_1, ...P_M\} \leftarrow \text{Retrieve}(P_{\text{init}}, \text{date}, \text{topic})$
    *# Stage 2: Table Extraction and Classification*
5: **for** each Leaderboard iteration $i = 1$ to $Iters$ **do**
6:     Count frequency of all datasets and retain top-K datasets from $U$ papers.
7:     **for** each dataset $j = 1$ to $K$ **do**
8:         Split $P_i$, Extract $U$ Tables $\{\text{Table}_1, ..., \text{Table}_U\}$ and table-related description $\{D_1, ...D_U\}$.
9:         Classify each table and keep "Main Results Table".
10:         **for** each main table and table description **do**
11:             Extract Paper title, Dataset, Metrics, Experiment Settings, and Experiment Results as quintuple.
12:         **end for**
        *# Stage 3: Leaderboard Generation*
13:         Recombine all quintuples, refine and rank the quintuples by performance scores.
14:         Output the Candidate Leaderboard $L_{ca}$
15:     **end for**
16: **end for**
    *# Stage 4: Quality Evaluation and Iteration*
17: Evaluate and select the best leaderboard $L_{\text{best}} \leftarrow \text{Evaluate}(L_{ca1}, L_{ca2}, ..., L_{caN})$
18: **Output:** Refined and evaluated leaderboard $L_{\text{best}}$

Table 5: Results of leaderboard quality generated by leveraging the official APIs provided by OpenReview (openreview-py) and the ACL Anthology (acl-anthology), we systematically harvested camera-ready PDFs from top-tier venues (NeurIPS, ICML, ICLR, ACL, EMNLP, NAACL).

| Items | Topic-related Quality | | Model | Speed$_{/s}$ | Content Quality | | | |
| | Recall | Precision | | | Coverage | Latest | Structure | Multiaspect |
|---|---|---|---|---|---|---|---|---|
| 5 | $82.36_{\pm9.42}$ | $80.77_{\pm10.05}$ | Qwen2.5-7B | 176.41 | $3.96_{\pm0.42}$ | $3.83_{\pm0.37}$ | $3.54_{\pm0.40}$ | 3.78 |
| | | | Qwen2.5-14B | 188.17 | $4.34_{\pm0.42}$ | $4.28_{\pm0.35}$ | $3.95_{\pm0.31}$ | 4.19 |
| | | | GPT-4o | 103.32 | $4.66_{\pm0.40}$ | $4.83_{\pm0.34}$ | $4.43_{\pm0.37}$ | 4.64 |
| | | | O1-preview | 117.22 | $4.70_{\pm0.42}$ | $4.85_{\pm0.65}$ | $4.51_{\pm0.31}$ | 4.69 |
| | | | Human Writing | 433 | 4.83 | 4.93 | 4.90 | 4.89 |
| 10 | $77.25_{\pm8.17}$ | $79.96_{\pm8.66}$ | Qwen2.5-7B | 199.76 | $3.63_{\pm0.47}$ | $3.50_{\pm0.55}$ | $4.09_{\pm0.47}$ | 3.74 |
| | | | Qwen2.5-14B | 207.39 | $4.12_{\pm0.45}$ | $3.83_{\pm0.46}$ | $3.97_{\pm0.32}$ | 3.97 |
| | | | GPT-4o | 113.43 | $4.72_{\pm0.41}$ | $4.53_{\pm0.35}$ | $4.72_{\pm0.35}$ | 4.66 |
| | | | O1-preview | 136.57 | $4.61_{\pm0.32}$ | $4.66_{\pm0.66}$ | $4.40_{\pm0.41}$ | 4.56 |
| | | | Human Writing | 780 | 4.82 | 4.67 | 4.70 | 4.73 |
| 15 | $70.21_{\pm7.54}$ | $73.90_{\pm6.83}$ | Qwen2.5-7B | 243.11 | $3.36_{\pm0.18}$ | $3.29_{\pm0.22}$ | $3.45_{\pm0.33}$ | 3.37 |
| | | | Qwen2.5-14B | 255.64 | $3.66_{\pm0.25}$ | $3.37_{\pm0.20}$ | $3.55_{\pm0.27}$ | 3.53 |
| | | | GPT-4o | 124.55 | $4.59_{\pm0.24}$ | $4.33_{\pm0.29}$ | $4.35_{\pm0.20}$ | 4.42 |
| | | | O1-preview | 137.80 | $4.35_{\pm0.44}$ | $4.16_{\pm0.23}$ | $4.30_{\pm0.29}$ | 4.27 |
| | | | Human Writing | 1176 | 4.70 | 4.53 | 4.51 | 4.58 |

Table 6: The prompts of the table extraction (w and w/o the table classification COT procedure) differs in the **[EXAMPLE JSON]**. It is detailed in the Table 8.

---

**\<instruction\>**
**You are an expert in summarizing and extracting key content from LaTeX-formatted academic papers on computers and artificial intelligence. Please output your reply in the following JSON format:**

**\<format\>[EXAMPLE JSON]\</format\>**

**Key points:**
- In the "selected table's core results", other models' results are of no concern and should be omitted.
- The table's header metrics should be the same as the evaluation metrics chosen.
- The number of items in the "classification of tables" dict should be equal to the "number of tables" int value. These two items help you to identify the main result tables better.
- All three items about the settings in the JSON output should correspond to the proposed method's best performance in the selected table.
- Sometimes in the selected table, the proposed method's performance may not be unique (e.g., different hyperparameters or training strategies); you need to choose the best one, which usually appears in the last row of the table.
- If there are multiple tables that meet the requirements (both being the main result table and based on the specified dataset), choose the one with richer information.

**Workflow example:**
First, I provide you with an article:

**\<article\>[EXAMPLE ARTICLE]\</article\>**

Then I specify the dataset as **[EXAMPLE DATASET]**, and you should output:

**\<format\>[EXAMPLE RESPONSE]\</format\>**

**\</instruction\>**

Table 7: Prompt of the leaderboard construction.

---

**<instruction>**
**You are an expert in constructing the Artificial Intelligence leaderboard. Please refer to the content I provide you to answer the user's questions. The contents I provide you are a number of structured summaries extracted from computer/artificial intelligence papers.**
**You need to build a markdown format leaderboard (showcase the performance of the models on the same dataset, each line representing a specific model) based on the titles, experimental settings, and evaluation metrics of these articles. Please output your reply in the Markdown format.**

Here, I list a complete example of the question and the answer to help you understand your task. For example, I provide you a list of JSON files containing the extracted content of the articles:

**[JSON LIST]**

The expected leaderboard that you generate should be:

**[EXAMPLE LEADERBOARD]**

———————————————— *Pay attention* ————————————————
- The leaderboard should be in the Markdown format and reflect all the articles provided!
- The leaderboard in the dictionary format is forbidden!
- In the above case, selecting Pre and Rec as the metrics in the final leaderboard is not appropriate because in most articles the corresponding performance values are absent.

Here, the target list of extracted content of the articles is as follows:

**[TARGET JSON LIST]**

**Warning:**
- **I need a well-organized markdown-format leaderboard containing all the articles' information. The leaderboard's max serial number in the "No." column should equal to the number of articles provided.**
- **When selecting metrics, you need to consider their text descriptions. The same metric may have multiple different abbreviations. In the final table, there must not be any duplicate metrics (it is unacceptable to have duplicates where different abbreviations represent the same meaning).**
- **Large-scale omissions are not allowed! For each model, only a small portion of the results are missing under the selected metrics. The vast majority of the metrics have corresponding values. The abbreviations for the same metric may be different, but you need to avoid being misled by the abbreviations.**
- **Use approximate intersections to select metrics from the given articles, while avoiding a large amount of data waste. Allow some models to have a certain degree of data missing under the selected metrics.**
- **The content in the "Experimental Setting" column should be concise and non-descriptive, just a few words.**
- **When different articles use different units for the same metric, please note that you need to convert them when integrating them so that the units in the final leaderboard are consistent. For example, 50% is equal to 0.5. "50" and "0.5" should not be presented in the same column of a leaderboard.**
- **Check each column corresponding to the selected metrics in the final leaderboard. If more than 60% of the values in that column are missing or represented by placeholders, the metric should be discarded.**

**</instruction>**

## C EXAMPLE PROMPTS

The prompts of instructing LLMs in different stages of League are illustrated in Box 6, and 7.

## Semi-Supervised Medical Image Segmentation Leaderboard: LA dataset

Papers due: 2024 December

Latest 20 papers

| No. | Model | | Experimental Setting | | Metrics | | | |
|---|---|---|---|---|---|---|---|---|
| | Model Name | Code | Training Strategy | | Dice | Jaccard | 95HD | ASD |
| 1 | **Uncertainty-Guided Cross Attention Ensemble Mean Teacher for Semi-supervised Medical Image Segmentation**
UG-CEMT framework with V-Net backbone
labeled data percentage of 20%, EWA decay rate of 0.99, dropout rate of 0.5, SAM radius of 0.5 | GitHub | semi-supervised learning with uncertainty-guided consistency regularization | | 89.73 | 81.63 | 2.2 | 0.5 |
| 2 | **Biologically-inspired Semi-supervised Semantic Segmentation for Biomedical Imaging**
UNet-like architecture
labeled data percentage of 20% | GitHub | two-stage semi-supervised approach | | 89.17 | 80.45 | 11.92 | 2.66 |
| 3 | **GraphCL: Graph-based Clustering for Semi-Supervised Medical Image Segmentation**
GraphCL with a 3D V-Net backbone
labeled scans of 8 (10%), unlabeled scans of 72, alpha of 0.5, kappa of 0.01, tau of 2 | - | Graph-based clustering with a teacher-student framework | | 90.24 | 82.31 | 6.42 | 1.71 |
| 4 | **Leveraging CORAL-Correlation Consistency Network for Semi-Supervised Left Atrium MRI Segmentation**
V-Net backbone
labeled scans of 16, unlabeled scans of 64, batch size of 4, learning rate of 0.01, momentum of 0.9, weight decay of 0.0001 | - | semi-supervised learning with CORAL-Correlation Consistency Network (CORN) | | 91.22 | 83.96 | 5.34 | 1.54 |
| 5 | **Dual-Teacher Ensemble Models with Double-Copy-Paste for 3D Semi-Supervised Medical Image Segmentation**
V-Net backbone
labeled_ratio of 20%, similarity_threshold of 0.01, EMA_decay_rate of 0.99 | GitHub | dual-teacher framework with staged selective ensemble and double-copy-paste strategy | | 91.82 | 84.92 | 5.11 | 1.5 |
| 6 | **Affinity-Graph-Guided Contractive Learning for Pretext-Free Medical Image Segmentation with Minimal Annotation**
Semi-AGCL framework
Labeled of 5%, Unlabeled of 95% | - | Affinity-graph-guided semi-supervised contrastive learning | | 90.44 | 79.05 | 7.78 | 2.11 |
| 7 | **Manifold-Aware Local Feature Modeling for Semi-Supervised Medical Image Segmentation**
V-Net architecture
alpha of 0.05 | GitHub | semi-supervised learning with 10% labeled data | | 90.28 | 82.37 | 6.49 | 1.66 |
| 8 | **SDCL: Students Discrepancy-Informed Correction Learning for Semi-supervised Medical Image Segmentation**
VNet and ResNet
labeled images of 8, unlabeled images of 72, batch size of 8, learning rate of 0.001, gamma of 0.5, mu of 0.05 | GitHub | semi-supervised learning with discrepancy correction learning | | 92.35 | 85.83 | 4.22 | 1.44 |
| 9 | **PMT: Progressive Mean Teacher via Exploring Temporal Consistency for Semi-Supervised Medical Image Segmentation**
V-Net
labeled percentage of 10%, EMA decay rate of 0.99, batch size of 4, iterations of 6000 | GitHub | Progressive Mean Teacher framework with pseudo-label filtering and discrepancy-driven alignment | | 90.81 | 83.23 | 5.61 | 1.5 |
| 10 | **Adaptive Mix for Semi-Supervised Medical Image Segmentation**
V-Net
labeled data percentage of 20%, mix-up patch size of 32, maximum number of mix-up patches of 16 | GitHub | AdaMix-MT framework (Mean-Teacher paradigm) | | 91.87 | 85.36 | 5.53 | 1.65 |
| 11 | **Self-Paced Sample Selection for Barely-Supervised Medical Image Segmentation**
SPSS framework with 16 labeled slices
learning rate of 0.01, iterations of 6000, decay of 0.1 every 2500 iterations | GitHub | self-paced sample selection framework with SU and SC components | | 86.19 | 75.89 | - | 3.49 |
| 12 | **Leveraging Task-Specific Knowledge from LLM for Semi-Supervised 3D Medical Image Segmentation**
V-Net backbone
labeled data percentage of 10%, unlabeled data percentage of 90% | - | co-training framework with unified segmentation loss | | 91.45 | 84.31 | 4.66 | 1.62 |
| 13 | **Rethinking Barely-Supervised Volumetric Medical Image Segmentation from an Unsupervised Domain Adaptation Perspective**
V-Net
labeled data percentage of 5% | GitHub | Barely-supervised learning via unsupervised domain adaptation (BvA) | | 87.4 | - | - | 2.37 |
| 14 | **Leveraging Fixed and Dynamic Pseudo-labels for Semi-supervised Medical Image Segmentation**
V-Net
labeled data ratio of 5%, unlabeled data ratio of 95% | - | co-training framework with fixed and dynamic pseudo-labels | | 89.55 | 81.18 | 5.48 | 1.99 |
| 15 | **CrossMatch: Enhance Semi-Supervised Medical Image Segmentation with Perturbation Strategies and Knowledge Distillation**
V-Net
labeled data percentage of 10%, confidence threshold (tau) of 0.85, distillation balance (eta) of 0.3 | GitHub | Self-training with knowledge distillation and perturbation strategies | | 91.33 | 84.11 | 5.29 | 1.53 |
| 16 | **Mixed Prototype Consistency Learning for Semi-supervised Medical Image Segmentation**
V-Net backbone
labeled scans of 16 (20%), unlabeled scans of 64 (80%), batch size of 4, learning rate of 0.01 | - | Mixed Prototype Consistency Learning framework with Mean Teacher and auxiliary network | | 91.98 | 85.02 | 4.77 | 1.58 |
| 17 | **An Evidential-enhanced Tri-Branch Consistency Learning Method for Semi-supervised Medical Image Segmentation**
ETC-Net with V-Net backbone
labeled scans of 8, unlabeled scans of 72, batch size of 4, learning rate of 0.1 | GitHub | semi-supervised learning with evidential tri-branch consistency | | 91.15 | 83.8 | 5.45 | 1.65 |
| 18 | **EPL: Evidential Prototype Learning for Semi-supervised Medical Image Segmentation**
V-Net architecture
learning rate of 0.001, batch size of 3, iterations of 10000 | - | semi-supervised learning with 20% labeled data | | 92.3 | 85.72 | 4.73 | 1.38 |
| 19 | **Uncertainty-aware Evidential Fusion-based Learning for Semi-supervised Medical Image Segmentation**
V-Net
labeled_ratio of 100%, unlabeled_ratio of 0% | - | semi-supervised learning with evidential fusion-based framework | | 92.62 | 85.24 | 4.47 | 1.33 |
| 20 | **Guidelines for Cerebrovascular Segmentation: Managing Imperfect Annotations in the context of Semi-Supervised Learning**
UA-MT (Uncertainty-Aware Mean-Teacher)
learning rate of 0.01, final weight for consistency loss of 0.01 | GitHub | semi-supervised learning with uncertainty-aware consistency regularization | | 89.51 | 81.01 | - | - |

Figure 5: A leaderboard (20 lines) of semi-supervised medical image segmentation on the LA dataset, using GPT-4o for table extraction and Qwen2.5-14B for leaderboard construction & refinement.

# Image Quality Assessment Leaderboard: LIVE dataset

**Papers due: 2024 November**

**Latest 20 papers**

| No. | Model Name | Code | Training Strategy | SROCC | PLCC |
|---|---|---|---|---|---|
| | **Model** | | **Experimental Setting** | **Metrics** | |
| 1 | **Dual-Representation Interaction Driven Image Quality Assessment with Restoration Assistance**
DRI-IQA model | GitHub | Dual-Representation Interaction method with restoration assistance | 0.982 | 0.984 |
| 2 | **Study of Subjective and Objective Quality in Super-Resolution Enhanced Broadcast Images on a Novel SR-IQA Dataset**
ARNIQA model | - | 5-fold cross-validation | 0.86 | 0.911 |
| 3 | **Exploring Rich Subjective Quality Information for Image Quality Assessment in the Wild**
RichIQA model with three-stage quality prediction network | - | multi-label training strategy using MOS, DOS, and SOS | 0.8943 | 0.9121 |
| 4 | **Q-Ground: Image Quality Grounding with Large Multi-modality Models**
Mask2Former | GitHub | semantic segmentation finetuning | - | - |
| 5 | **Dual-Branch Network for Portrait Image Quality Assessment**
Dual-Branch Network with Swin Transformer-B backbones | GitHub | Pre-trained on LSVQ and GFIQA datasets, followed by learning-to-rank optimization | 0.85 | 0.86 |
| 6 | **Cross-IQA: Unsupervised Learning for Image Quality Assessment**
ViT (Vision Transformer) with Cross-IQA pretraining | - | unsupervised pretraining followed by fine-tuning | 0.965 | 0.976 |
| 7 | **Deep Bi-directional Attention Network for Image Super-Resolution Quality Assessment**
BiAtten-Net | GitHub | Bi-directional attention mechanism for full-reference IQA | 0.981 | 0.982 |
| 8 | **High Resolution Image Quality Database**
HR-BIQA model with modified ResNet50 and ViT | GitHub | patch-based BIQA model designed for high-resolution images | 0.92 | 0.925 |
| 9 | **Deep Shape-Texture Statistics for Completely Blind Image Quality Evaluation**
EfficientNet-b7 | - | Shape-Texture Adaptive Fusion (STAF) module with shape and texture CNN branches | 0.935 | 0.931 |
| 10 | **JOINT DEEP IMAGE RESTORATION AND UNSUPERVISED QUALITY ASSESSMENT**
QAIRN (Quality-Aware Image Restoration Network) | - | Joint restoration and unsupervised quality assessment | 0.879 | 0.87 |
| 11 | **Perceptual Assessment and Optimization of HDR Image Rendering**
HDR-NeRF with multilayer perceptron (MLP) | GitHub | Perceptual optimization using HDR quality metrics | 0.869 | 0.873 |
| 12 | **Blind Image Quality Assessment via Transformer Predicted Error Map and Perceptual Quality Token**
ViT-B/16 (Vision Transformer backbone) | GitHub | Pre-training on KADID-10K dataset followed by fine-tuning on LIVE dataset | 0.976 | 0.977 |
| 13 | **Gap-closing Matters: Perceptual Quality Evaluation and Optimization of Low-Light Image Enhancement**
IACA (Illumination Aware and Content Adaptive model) | GitHub | Deep learning-based IQA model trained on SQUARE-LOL database | 0.875 | 0.878 |
| 14 | **Explainable Image Quality Assessments in Teledermatological Photography**
EfficientNet-B0, 15 MB | - | supervised learning with class-weighted training | - | - |
| 15 | **Image Quality Assessment with Gradient Siamese Network**
Gradient Siamese Network (GSN) | - | Trained on the entire KADID-10k dataset and tested on LIVE dataset | 0.932 | 0.922 |
| 16 | **DeepWSD: Projecting Degradations in Perceptual Space to Wasserstein Distance in Deep Feature Space**
DeepWSD with VGG16 backbone | GitHub | No training with quality labels, pre-trained network | 0.9624 | 0.9609 |
| 17 | **Perceptual Quality Assessment for Fine-Grained Compressed Images**
Proposed method with gradient-based and texture-based features | - | Full-reference image quality assessment (FR-IQA) method | 0.973 | 0.9612 |
| 18 | **SPQE: Structure-and-Perception-Based Quality Evaluation for Image Super-Resolution**
SPQE metric with HR as reference | - | end-to-end training with adaptive tradeoff mechanism | 0.9317 | 0.9641 |
| 19 | **Multi-Scale Features and Parallel Transformers Based Image Quality Assessment**
MSFPT-avg (Multi-Scale Features and Parallel Transformers) | GitHub | Full-Reference IQA with multi-scale feature extraction and parallel transformers | 0.977 | 0.972 |
| 20 | **Content-Variant Reference Image Quality Assessment via Knowledge Distillation**
CVRKD-IQA with FR-teacher | GitHub | Knowledge distillation from FR-teacher to NAR-student | 0.973 | 0.969 |

Figure 6: A leaderboard (20 lines) of image quality assessment on the LIVE dataset, using GPT-4o for both table extraction and leaderboard construction & refinement.

# Image Quality Assessment Leaderboard: LIVE dataset

**Papers due: 2024 November**

**Latest 5 papers**

| No. | Model Name | Code | Training Strategy | SROCC | PLCC | RMSE | mIoU | mAcc |
|---|---|---|---|---|---|---|---|---|
| 1 | **Dual-Representation Interaction Driven Image Quality Assessment with Restoration Assistance** 
 DRI-IQA model 
 learning rate of 2e-4, batch size of 64 | GitHub | Dual-Representation Interaction method with restoration assistance | 0.982 | 0.984 | - | - | - |
| 2 | **Study of Subjective and Objective Quality in Super-Resolution Enhanced Broadcast Images on a Novel SR-IQA Dataset** 
 ARNIQA model 
 scaling factor x2, iterations 1000 | - | 5-fold cross-validation | 0.86 | 0.911 | 0.699 | - | - |
| 3 | **Exploring Rich Subjective Quality Information for Image Quality Assessment in the Wild** 
 RichIQA model with three-stage quality prediction network 
 Adam optimizer with an initial learning rate of 0.00001, batch size of 8 | - | multi-label training strategy using MOS, DOS, and SOS | 0.8943 | 0.9121 | 8.2312 | - | - |
| 4 | **Q-Ground: Image Quality Grounding with Large Multi-modality Models** 
 Mask2Former 
 learning rate of 0.0003, batch size of 2 | GitHub | semantic segmentation finetuning | - | - | - | 0.403 | 0.646 |
| 5 | **Dual-Branch Network for Portrait Image Quality Assessment** 
 Dual-Branch Network with Swin Transformer-B backbones 
 Initial learning rate of 1e-5, batch size of 12 | GitHub | Pre-trained on LSVQ and GFIQA datasets, followed by learning-to-rank optimization | 0.85 | 0.86 | - | - | - |

Figure 7: A leaderboard (5 lines) of image quality assessment on the LIVE dataset, using GPT4-o for both table extraction and leaderboard construction & refinement.

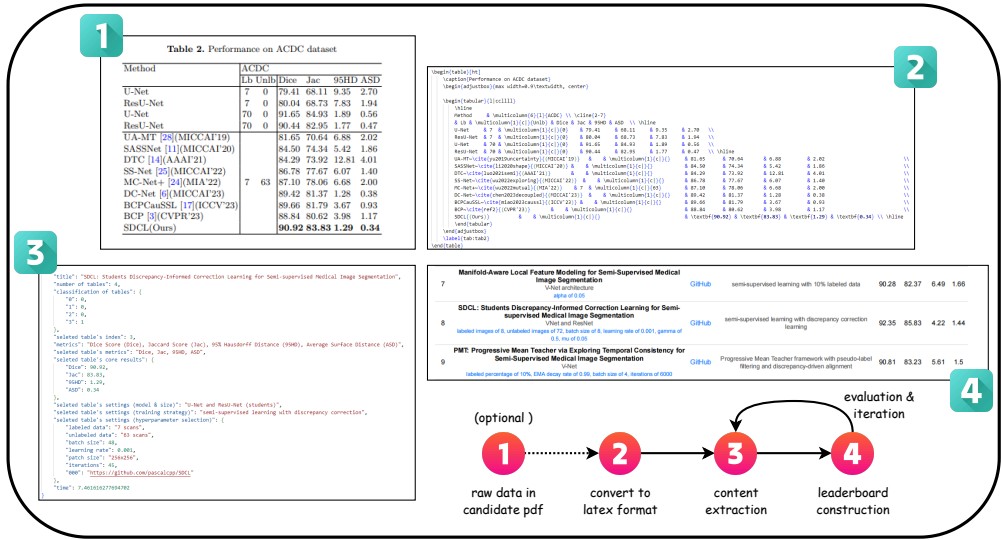

Figure 8: The workflow of League from a single table's perspective.

Table 8: The example JSON file of the table extraction with table classification COT.

```
1  {
2      "title": "The title of the paper (String)",
3      "number of tables": "The number of tables in the paper,
          ↪ denoted as n (Int)",
4      "classification of tables": {
5      "0": "main-result/comparison",
6      "1": "ablation",
7      "2": "hyper-parameter",
8      "3": "others"
9      },
10     "selected table's index": "The index of the main result table
          ↪ focused on the specified dataset [SPECIFIED DATASET],
          ↪ denoted as i (Int)",
11     "metrics": "The evaluation metrics chosen to assess
          ↪ performance of the method proposed in this paper. This
          ↪ information is extracted from the textual portion of the
          ↪ 'Experimental' related section (String)",
12     "selected table's metrics": "Metrics used in the selected main
          ↪  result table, it should be almost the same as the
          ↪ metrics extracted from the textual. Remove the latex
          ↪ format syntax (String)",
13     "selected table's core results": "A dictionary only containing
          ↪  this paper's model best performance on the selected
          ↪ dataset, with the metrics as keys and the corresponding
          ↪ values (Dict)",
14     "selected table's settings (model & size)": "In computer
          ↪ vision, the model usually means the backbone architecture
          ↪  of the network, such as ResNet, ViT, and so on. The size
          ↪  can be omitted if not specified. In NLP, the model and
          ↪ size are usually organized as a string, such as 'LLAMA-7B
          ↪ ', 'GPT-3', and so on (String)",
15     "selected table's settings (training strategy)": "Training
          ↪ strategy usually refers to the concepts like: fine-tuning
          ↪ , transfer learning, linear-probing, reinforce learning,
          ↪ one-shot, few-shot, prompt-learning, semi/self supervised
          ↪  and so on (String)",
16     "selected table's settings (hyperparameter selection)": "The
          ↪ hyperparameters used in the model, such as learning rate,
          ↪  batch size, and so on. You should output a dictionary
          ↪ with the hyperparameters and their values (Dict)",
17     "github": "The link to the gitHub repository containing the
          ↪ code for this paper, if available (String)"
18  }
```

## D   IMPLEMENATION DETAILS

**Table Classification and Table NER**   To illustrate the effectiveness of stages 2 and 3. We manu-
ally annotated 354 tables from 72 papers, with 197 main results tables, 54 ablation study tables, and
103 others. For Table NER, we follow the annotation criteria of SciIE (Li et al., 2023) and anno-
tated 336 entities from 28 tables. The entities contain 172 methods, 31 datasets, 49 settings, and 84
metrics. As illustrated in Table 3, we using different prompt strategies: with 0-shot and 1-shot. For
prompt with 0-shot, we just provide a brief definition of the sub-categories. For 1-shot, we give an
example table for each category besides the definition.

**Leaderboard Construction**   For proprietary models, we employ official APIs to interact with ex-
clusive LLMs, and the prompts are well-defined. we set temperature = 0.3 and other parameters as

Table 9: Leaderboard Quality Criteria.

| Criteria | Scores |
|---|---|
| **Coverage** | The ratio of the number of papers used for leaderboard generation to the total number of papers searched. $(P_{used}/P_{total}) * 5$ |
| **Latest** | The ratio of the number of papers published after the certain date to the total number of papers searched. $(P_{new}/P_{total}) * 5$ |
| **Structure** | *Score 1*: The structure of the leaderboard lacks logic, making it difficult to understand and navigate. The table header and each row are not clearly organized and connected. 
 *Score 2*: The structure of the leaderboard have some contents arranged in a disordered or unreasonable manner. However, the overall structure is reasonable and coherent. 
 *Score 3*: The survey is generally comprehensive in coverage but still misses a few key points that are not fully discussed. 
 *Score 4*: The structure of the leaderboard is generally reasonably logical, with most header items arranged orderly, though some header items may be repeated or redundant. 
 *Score 5*: The structure of the leaderboard has good logical consistency, with each line strictly related to the header items and the previous line. But it can be optimized in terms of easy understanding. |
| **Multi-Aspect** | The evaluation metric for multi-leaderboard. Specifically, a research topic $T$ may have $N$ different datasets, and thus we can get $N$ leaderboards, the score of the Multi-Aspect is computed based on the average of all the $N$ scores. $(N_{Coverage} + N_{Latest} + N_{Structure})/(3 * N)$. |

Table 10: Illustration of the table's cell entity recognition, where ☐ stands for datasets, ☐ for methods, ☐ for metrics, and ☐ for experimental settings.

| | KonIQ-10K | |
|---|---|---|
| Method | SRCC↑ | PLCC↑ |
| w/o direct pathway | 0.9376 | 0.9495 |
| w/o indirect pathway | 0.9361 | 0.9479 |
| w/o both pathways | 0.9363 | 0.9463 |
| **RichIQA** | **0.9383** | **0.9500** |

default. For open-source models, all experiments are conducted on a single A100 GPU. The input length is set to 128K tokens and the max output tokens is 4096.

# E  EXAMPLE LEAGUE-GENERATED LEADERBOARDS

Figure 5 and 6 illustrate two examples generated by League. The papers in the first leaderboard are the latest methods of semi-supervised medical image segmentation on the LA dataset from March to December in 2024. The second leaderboard collects the most recent methods for image quality assessment conducted on the LIVE dataset from February 2022 to November 2024. To ensure that the table content is fully displayed, the "model & size" and "hyperparameters selection" within the experimental settings are presented beneath the paper titles.

First and foremost, when viewed holistically, both leaderboards with 20 entries, whether utilizing qwen2.5-14B or GPT-4o as the construction model, exhibit a notably high level of completeness. Upon specific analysis of the missing information, in Leaderboard 1, League failed to successfully extract the HD value from "Self-Paced Sample Selection for Barely-Supervised Medical Image Segmentation" (No. 11) because the metric was referred to as 95HD in the original text. Although our design accounts for such situations, required to extract metrics from both text and tables to avoid confusion caused by abbreviations. This design has successfully resolved most of the issues arising from abbreviations, but such errors still occur with a small probability. The absence of metrics in entries No. 13 and No. 20 is acceptable because the original text indeed lacks these metrics. The situation in Leaderboard 2 is similar; the only two missing items (No. 4 and No. 14) are also due to the absence of corresponding results in the original texts.

The higher missing rate in the 5-row leaderboard compared to the 20-row leaderboard for the LIVE dataset can be attributed to the following reasons: When only 5 papers are included, League extracts a larger number of metrics, including RMSE, mIoU, and mAcc. The missing values for these metrics are tolerable in a 5-row leaderboard. However, when expanding to a 20-row leaderboard,

Table 11: Cost of API. The open-source Qwen2.5-7/14B model is deployed on a local server comprising four A800 GPUs, resulting in zero cost.

| Input tokens | Output tokens | Qwen2.5-7/14B | kimiAI-128k | GPT4-o | O1-preview |
|---|---|---|---|---|---|
| 834.7K | 8.9K | 0 $ | 7.034 $ | 2.176 $ | 13.055 $ |

the excessive number of missing values forces League to discard these metrics to ensure that the leaderboard conveys meaningful information.

Secondly, regarding the experimental settings, we observe that in Leaderboard 1, the information on "model & size", "hyperparameters", and "training strategy" is both accurate and comprehensive. Notably, there is a consistent thread throughout the hyperparameters: the portion of labeled data. In contrast, Leaderboard 2 discards the hyperparameter information compared to Leaderboard 3. This is because we require League to extract hyperparameter information in a way that not only maintains completeness but also focuses on the intrinsic connections between different items. If the deviation is too large (i.e., if it cannot provide users with a concise and effective summary), the information should be discarded. Therefore, when the number of input papers for League increases from 5 to 20, the hyperparameter settings in the topic of image quality assessment do not have a clear and unified theme and thus are ultimately ignored.

## F    WORKFLOW ILLUSTRATION

Figure 8 illustrates the transformation of the main experimental table of the LA dataset in the SDCL (Song & Wang, 2024) through the entire workflow of League. Initially, the table information is presented in the form of visual features within a PDF file. Subsequently, through the processes of crawling and LaTeX integration, the table is extracted and classified into an independent LaTeX format. Further, the table information is structured by the COT table extraction. Finally, after evaluation and iteration, it becomes an entry in the final leaderboard.

## G    COST ANALYSIS

We calculate the average number of input & output tokens required to generate a 20-entry leaderboard, along with the cost analysis using different LLMs, as shown in Table 11. The computational cost of all models remains within 14$, indicating that League is also economically efficient. Overall, the League framework consumes more input tokens, while the output tokens represent only a small proportion. OpenAI prices output tokens significantly higher than input tokens, often reaching 4-5 times the cost of input tokens. However, considering the disparity in token numbers, the overall cost remains acceptable.

