# OpenReview forum: "League: Leaderboard Generation on Demand"
_ICLR.cc/2026/Conference — ICLR 2026 Conference Withdrawn Submission_

### Official Review · Reviewer_Nh2B · 2025-10-30

**Soundness:** 3
**Presentation:** 3
**Contribution:** 2
**Rating:** 2
**Confidence:** 4

**Summary:**

This paper introduces Leaderboard Auto Generation (League), a structured framework for automatically generating leaderboards for a given AI research topic. League consists of four key stages: (1) paper collection, (2) result extraction and integration, (3) leaderboard generation, and (4) quality evaluation. The authors apply League to generate leaderboards and assess them in terms of topic relevance, content quality, and construction efficiency.

**Strengths:**

- Reasonable and well-designed pipeline: League presents a clear and structured approach to leaderboard construction. The four-stage design is intuitive and logically organized.
- Ablations and stage-wise validation: Intermediate evaluations and ablation studies effectively demonstrate the contribution and effectiveness of each stage and strategy within League.

**Weaknesses:**

- Lack of baselines: The paper only compares League against human-written leaderboards. It does not include comparisons with prior automated or semi-automated methods listed in Table 1, which weakens the empirical evaluation.
- Questionable evaluation: The size of the evaluation set is not clearly described (Table 2). More details are needed on how the evalaution set is collected and categorized. Additionally, the use of LLM-as-a-judge for these metrics raises concerns (see questions below).

**Questions:**

- Did you assess the agreement between LLM-as-a-judge and human evaluators? What is the consistency rate, and can LLMs reliably serve as fair judges for leaderboard quality evaluation?
- Could you provide more details on the evaluation dataset? How were the evaluation samples collected and categorized?

---

### Official Review · Reviewer_LAsN · 2025-10-31

**Soundness:** 3
**Presentation:** 3
**Contribution:** 2
**Rating:** 4
**Confidence:** 3

**Summary:**

This paper introduces League, an end-to-end framework for automatically generating dynamic leaderboards on rapidly evolving research topics using large language models (LLMs). League automates the entire process from paper collection and result extraction to leaderboard generation and evaluation achieving quality comparable to manual curation with minimal human effort. Unlike prior works that focus mainly on static entity extraction or limited paper snapshots, League emphasizes continuous and fair leaderboard construction by dynamically tracking, aligning, and comparing model performance across datasets and metrics under standardized evaluation settings. Key challenges addressed include limited paper coverage, inconsistent comparisons, and low timeliness in existing methods.
League’s pipeline consists of four stages: (1) Paper Collection: Relevant LaTeX and PDF files are retrieved from arXiv and top-tier conferences using retrieval models and filtered by date and topic. (2) Table Extraction and Classification: LLMs extract and classify experimental tables (main results, ablations, or others) through in-context learning; only “main result” tables are retained and extracted by chain of thought reasoning. (3) Table Unpacking and Integration: Core experimental details (datasets, metrics, settings, results) are extracted, and datasets used in at least five papers are selected for leaderboard inclusion. (4) Leaderboard Generation and Evaluation: Extracted results are refined, re-ranked, and integrated into comprehensive leaderboards.

**Strengths:**

- The paper presents a clear and well-motivated problem: the difficulty of maintaining up-to-date leaderboards amid the rapid growth of research papers.
- The proposed framework is systematic and robust, providing an end-to-end solution covering paper collection, result extraction, and leaderboard generation.
- It achieves high efficiency, significantly reducing the time and effort compared to manual curation.

**Weaknesses:**

- The framework is well engineered but mainly extends existing LLM-based pipelines without introducing fundamentally new algorithms or modeling techniques.
- While structure and coverage quality improve across iterations, the latest score remains largely unchanged. The paper needs deeper analysis of how iterative refinement improves leaderboard quality, especially during the refinement stage.
- The paper claims to ensure fair comparisons by aligning experiment settings but does not specify quantitative thresholds for excluding hyperparameter deviations, making the fairness criterion somewhat subjective.

**Questions:**

- Q1) Topic-Related Quality Evaluation

While content quality is assessed via LLM-as-Judge, how are recall and precision for topic-related quality computed? What serves as the ground-truth reference for determining whether a paper or leaderboard item is “related”?

- Q2) Iteration and Latest Score Mechanism

Since papers are filtered by date in stage 1, how does the subsequent refinement and re-evaluation process in Stage 4 specifically contribute to the improved latest score? Does the iteration process help prioritize or re-rank newer results that might have been initially marginalized?

- Q3) Handling Conflicting or Multiple Experimental Settings

If a paper reports multiple sets of results for the same model/dataset using significantly different experimental settings (e.g., different regularization methods or optimization algorithms) within the main results table, how does League select the single, representative result/setting to form the quintuple, especially when the goal is to capture the best performance?

---

### Official Review · Reviewer_2dKC · 2025-10-31

**Soundness:** 2
**Presentation:** 2
**Contribution:** 1
**Rating:** 2
**Confidence:** 5

**Summary:**

This paper presents a new task, leaderboard construction, along with a benchmark and evaluation of several versions of their system.
The task is important to the AI community, which is inundated with benchmarks that are not updated.

**Strengths:**

The primary strength of this paper is the importance of the task to the practice of AI. A high quality solution to this problem would have a significant impact on the field.
I appreciate that the authors performed some analysis on different versions of their system, and performed some "meta evaluation" measurement on correlation of their automatic metrics with human judgements.

**Weaknesses:**

- The "internal validity" of the benchmark is questionable due to the very limited meta evaluation. The authors only performed a limited pairwise annotation task with humans and looked at correlation with automatic metrics. The automatic metrics are not well established, and would have expected a much more significant study to validate them, or establish a proper ground truth. This is especially important for this work as I view the primary contribution here being the new task formulation.
- The paper would be stronger if it justified why this is an important task to study beyond the significance of the task to the practice of AI---for example by characterizing more crisply how this problem sheds new light on LLM capabilities beyond existing benchmarks.
- The paper also provides an overly broad motivation that it does not deliver on, and does not deliver on solving the problems of the lack of control or reproducibility of benchmark experiments---for example, it does not extract and validate hyperparameters or validate scores by executing code.

**Questions:**

- I would be interested to hear responses to the weaknesses stated above; for example, could the authors shed more light on the validity of the automatic metrics?
- Figure 1 right is illegible

---

### Official Review · Reviewer_rApG · 2025-11-03

**Soundness:** 2
**Presentation:** 3
**Contribution:** 2
**Rating:** 4
**Confidence:** 3

**Summary:**

This paper proposes League, a four-stage, end-to-end pipeline that automatically constructs topic-specific research leaderboards: (1) paper collection and filtering (arXiv + top venues), (2) table extraction and classification, (3) table unpacking into quintuples (title, dataset, metrics, settings, results), and (4) leaderboard recombination, refinement, and LLM-as-judge evaluation. Experiments report topic relevance (precision/recall), content quality judged by LLMs and calibrated with human ratings, speed vs. manual curation, ablations (no table-classification / no refinement), and an example HotpotQA leaderboard that is more up-to-date than the public one. Across 5–20 item leaderboards, League approaches human quality while being ~10× faster, and claims to be the first to move beyond entity extraction to dynamic leaderboard generation with settings-aware comparisons.

**Strengths:**

* Clear, modular pipeline with practical scope. The four stages and their iteration are well specified, including the choice to operate on LaTeX sections/tables and to retain only “main results”. This design choice addresses token cost and noise pragmatically.

* Settings-aware comparison. Extracting experiment settings (e.g., model size, data size) alongside metrics to support fairer comparisons goes beyond prior T-D-M extraction and is valuable if robust.

* Comprehensive evaluation framing. The paper separates Topic-related Quality (precision/recall of relevance) from Content Quality (Coverage, Latest, Structure) and adds speed comparisons vs. manual curation. It also reports ablations that show table classification and refinement both help.

* Human–LLM agreement analysis. A meta-evaluation computes Pearson correlations between LLM-judged and human-judged leaderboard quality, with moderate-to-strong correlation (e.g., up to 0.76), providing some external validity for the LLM-as-judge protocol.

**Weaknesses:**

W1. The evaluation framework relies heavily on LLM-as-judge with limited verified ground truth. Topic-related precision and recall appear to depend on internally defined labels of “relevant” items, yet the paper does not fully specify the gold-standard construction, inter-annotator agreement, or adjudication of borderline topic matches. Content quality is primarily judged by LLMs, with only a correlation study to human preferences, introducing risks of circularity and sensitivity to prompt or model choices.

W2. The claim of “fair comparison” is only partially substantiated. Although the pipeline extracts evaluation settings, it remains unclear how incompatibilities (such as differing compute budgets, training data regimes, dev/test splits, reporting conventions, or metric variants) are normalized or controlled in the final rankings, particularly when multiple variants per paper exist. The warning in the authors' leaderboard-construction prompt about harmonizing units (Table 7) hints at fragility.

W3. The retrieval and coverage pipeline faces domain-shift and recall limitations. Stage 1 relies on regex or keyword filtering combined with a learned retriever over titles and abstracts, but many impactful results appear only in appendices, project pages, or workshop updates with non-obvious titles. While top-venue harvesting (OpenReview, ACL Anthology) enhances reliability, it risks omitting strong arXiv-only or late-breaking work; conversely, missing LaTeX sources in camera-ready PDFs can impede accurate table extraction.

W4. The example leaderboard contents exhibit heterogeneity and potential noise.
In the HotpotQA example (Figure 3), some rows mix different model families and training strategies, and several entries appear only loosely related to the core evaluation setup (e.g., inner-representation or hallucination metrics vs. EM/F1 multi-hop QA). It remains unclear whether all entries correspond to standard evaluations on the official HotpotQA test set under a consistent distractor configuration.

W5. The paper offers limited error and robustness analysis. Although the authors report averages by leaderboard length and ablations, there is no systematic examination of failure cases such as mis-classified tables, mis-parsed metrics, incorrect dataset disambiguation (e.g., similarly named benchmarks), or hallucinated fields.

**Questions:**

Please address all issues raised in Weaknesses.

Additional Questions for the Authors

Q1. Clarification of gold labeling and evaluation reliability. Please elaborate on the gold labeling protocol for topic relevance, including who performed the annotations, what guidelines were followed, and how IAA was measured. To support reproducibility, consider releasing the labeled sets or at least providing detailed annotation statistics and adjudication procedures.

Q2. Normalization of evaluation settings and metric variants. How are metric variants (e.g., EM vs. EM@k, macro vs. micro F1) and hidden protocol differences reconciled across sources? Please describe the detection and normalization process for such inconsistencies, including how data contamination (e.g., training on test-like corpora) is identified and handled. A quantitative audit of normalization or conversion errors would be valuable for assessing ranking reliability.

Q3. Retrieval coverage and data completeness. Please report recall@paper against a manually curated oracle list for several representative topics, in addition to the item-level recall currently presented. Quantify the proportion of table extraction failures on PDF-only papers and discuss how such cases affect overall coverage and bias in leaderboard construction.

---

### Note · Authors · 2025-11-13

I have read and agree with the venue's withdrawal policy on behalf of myself and my co-authors.